

# 1 Sensitivity study of the tropical Pacific precipitation
# 2 anomalies

Shouwen Zhang[1], Hua Jiang[1], Hui Wang[1,2], Ling Du[3], Dakui Wang[1]
[1]National Marine Environment Forecasting Center, State Oceanic Administration, Beijing 100081, China
[2]Key Laboratory of Research on Marine Hazards Forecasting, National Marine Environmental Forecasting
Center, Beijing 100081, China
[3]Department of Oceanography, Ocean University of China, Qingdao 266100, China
*Correspondence to*: Hua Jiang (hjiang@nmefc.gov.cn)
**Abstract.** Climate model results have shown that precipitation in the tropical Pacific Ocean will change
up to 15% and 25% in one century. In this paper, both reanalysis data and climate model are used to study
the response of global ocean and atmosphere to precipitation anomalies in the tropical Pacific Ocean. It
shows that positive precipitation anomalies could trigger an El Nino-like SSTA response, with warmer
SST in the east tropical Pacific Ocean and slightly cooler SST in the west tropical Pacific Ocean. The
zonal tropical ocean currents change significantly, of which the magnitudes and directions are mainly
relying on the intensity of the precipitation anomalies. Through a wave train encompassing the whole
Northern Hemisphere named as the Circumglobal Waveguide Pattern (CWP), the North Atlantic
atmospheric circulation responds to the freshwater anomalies in a NAO-like pattern. The anomalous
atmospheric circulation transport sea ice to the North Atlantic Ocean. The sea ice melts in summer and
freshen the upper ocean, which makes the ocean more stable. It thus constrains vertical heat transport and
makes the upper water cooler, forming a significant positive feedback mechanism.

**Key words:** precipitation; tropical Pacific Ocean; sensitivity; anomalies

## 25 1 Introduction

Air-sea freshwater flux, defined as precipitation (P) minus evaporation (E), (PmE), is an important
indicator representing the water exchange between atmospheric and ocean. Previous studies have found
that mean PmE would increase under global warming mainly through the increasing saturation level of
moisture, and 16% to 24% intensification of the global water cycle will occur in a future 2°to 3°warmer
world (Durack et al. 2012). Especially, the precipitation will experience significant changes in frequency
and spatial distribution in the future. Therefore, freshwater changes in response to climate change pose a
more severe risk to human societies and ecosystems than warming alone.
Prior studies mainly focus on the response of Atlantic thermohaline circulation to freshwater anomalies in
the North Atlantic high latitudes region, which usually take the water-hosing experiments as typical
research method and have reached qualitative conclusions (Broecker et al., 2003; Curry et al., 2003;
Timmermann et al., 2005; Wu et al., 2008). It has also been found that freshwater forcing plays an active
role in maintain the Pacific climate, especially the ENSO phenomenon (Huang and Mehta, 2005; Zhang et
al., 2009; Zheng et al., 2012). In addition, oceanic and atmospheric responses to local freshwater forcing
have been paid substantial attention (Williams et al., 2006; Stouffer et al., 2007; Zhang et al., 2011a,





2011b).
The moisture saturation will increase in a future persistent global warming world, resulting in significant
change of freshwater flux. Compared to evaporation, precipitation plays a more important role in the
freshwater change. The precipitation in the tropical Pacific Ocean will increase substantially over the
coming one hundred years (Laine et al., 2014). To quantitate the change, we analyzed the precipitation
variations following Laine et al., (2014) in Representative Concentration Pathway (RCP) 4.5 and 8.5
scenarios. Results have shown that the precipitation will change most significantly in the tropical Pacific
Ocean, exceeding 15% and 25% than their climatology in RCP 4.5 and 8.5 scenarios respectively in one
century. Meanwhile, the change of evaporation is comparatively weak, with an increase less than 5%. At
present, there are sufficient useful tools to research the effect of freshwater on the climate change owing to
the availability of the reliable oceanic and atmospheric datasets and the development of the global climate
models.
In this paper, historical oceanic and atmospheric datasets are used to evaluate the characteristics of ocean
currents and sea surface temperature (SST) during the strong precipitation periods using composite
analysis method. Then, experiment performed by the CESM model is used to study the response of global
ocean and atmosphere to a 10% increase of precipitation in the tropical Pacific Ocean. The arrangement of
the article is as follows, section 1 describes the research background and contents. Section 2 describes the
data and model used in this study. Section 3 analyzes the response of ocean currents and SST in the
extreme tropical Pacific Ocean precipitation context. The results of model simulation are shown in section
4. Finally, these results are summarized and discussed in section 5.
**2 Data and model**
The Global Precipitation Climatology Project (GPCP) dataset (Huffman et al., 1997; Adler et al., 2003) is
used as the source of precipitation data. To analyze the characteristics of tropical currents during extreme
tropical precipitation period, three datasets with the same period from 1980 to 2008 are used, which are
the European Centre for Medium-Range Weather Forecasts (ECMWF) Reanalysis datasets (ORA-S3,
Balmaseda et al., 2008), Global Ocean Data Assimilation System (GODAS, Behringer et al., 2004) and
Simple Ocean Data Assimilation (SODA) datasets. It's noteworthy that SST is also extracted from SODA
datasets.
Tropical Pacific Ocean (5 °S-5 °N, 120 °E-80 °W) mean precipitation is calculated first, and then 20 months
corresponding to the largest and smallest precipitation are obtained respectively. A composite analysis
method is carried out to examine the differences of SST and currents between strong and weak
precipitation over the tropical Pacific Ocean. In each of the analyses, a test of statistical significance based
on t-test, is performed to identify geographical regions where the composite results at the 95 % confidence
level.
We use the Community Earth System Model (CESM) version 1, a fully coupled, global climate model
maintained by the Climate and Global Dynamics Division (CGD) at the National Center for Atmospheric
Research (NCAR). CESM consists of five geophysical models: the atmosphere model is Community
Atmosphere Model (CAM5), the ocean model is based on the Parallel Ocean Program version 2
developed by the Los Alamos National Laboratory (POP2), the sea ice model is Community Ice Code
(CICE4), the land model is Community Land Model (CLM4) and the land ice model is Community Ice
Sheet Model (CISM). A coupler (CPL7) is used to coordinate the models and passe information among
them. The CESM used in this paper has an atmosphere resolution of 1.9(zonal)*2.5(meridional)*40, the





ocean model has 384(latitude)*320(longitude) grid points with 60 vertical levels (Danabasoglu et al.
2012). The sea ice model uses the same horizontal grid as that of the ocean model, while grids of the land
model and the atmosphere model are identical. Good performance in simulating the freshwater forcing and
climate change have been shown by CESM in many studies (Kirtman et al., 2011;Yeager et al., 2012;
Danabasoglu et al., 2012).
A control experiment integrated for 150 years is carried out focusing on the ocean surface and atmosphere
adjustment. Based on the precipitation projection in one century that the tropical Pacific precipitation will
change 15%~25%, we are conservative to add 10% of precipitation on each time-step in the coupler,
which has the same pattern of the most robust precipitation in the tropical Pacific Ocean (5°S-5°N,
120°E-80°W). The sensitivity experiment begins at the end of 60 years of the control experiment, and then
integrates forward for another 90 years. The difference between the sensitive experiment and the control
experiment is taken as the response, and the last 10 years of both the sensitivity experiment and the control
experiment are taken as the equilibrium response.
**3 Observation results**
Precipitation correlates significantly with SST over the tropical Pacific Ocean. As shown in Fig.1, SST
rises more than 6°C in the tropical eastern Pacific Ocean on account of the positive precipitation
anomalies of the whole tropical Pacific Ocean. It is accompanied by weak decreasing in the tropical
western Pacific Ocean, which forms an obvious temperature gradient over the tropical Pacific Ocean. The
spatial distribution of temperature anomalies caused by strong precipitation resembles with the SST
anomalies characteristics during El Nino periods.
In addition, equatorial currents also demonstrate significant changing. We use three different kinds of
current datasets to increase the credibility of the results, as the accuracy of ocean current datasets is much
lower than that of SST. It is shown that, three ocean current datasets have good manifestation in terms of
both location and magnitude. It's noteworthy that the north branch of the south equatorial current at 180°
in SODA datasets is much deeper than that in the other two datasets. Specifically, the south and north
equatorial currents show positive anomalies, especially for the north part of the south equatorial current,
which indicate strong weakening of the currents. For both profiles, the equatorial counter current shows
strong negative anomalies. The change is most significant as the anomaly is even larger than its
climatology. It's mainly because the equatorial counter current locates at the equatorial calm zone and
rainfall in this region is abundant. Changes of equatorial latent current of two profiles are slightly different.
Negative anomalies are robust in the middle and upper part of the 180° profile, while it is in the middle
and lower part of the 220° profile. The range of significant weakening could be up to half of its
climatology.



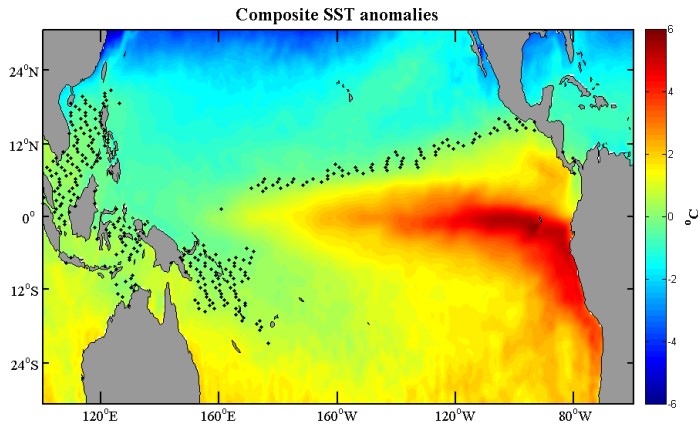

**Figure 1: Composite map of SST anomalies difference between strong and weak precipitation periods over the**
**tropical Pacific. Dotted regions are not at 95% confidence level. Units: ℃**

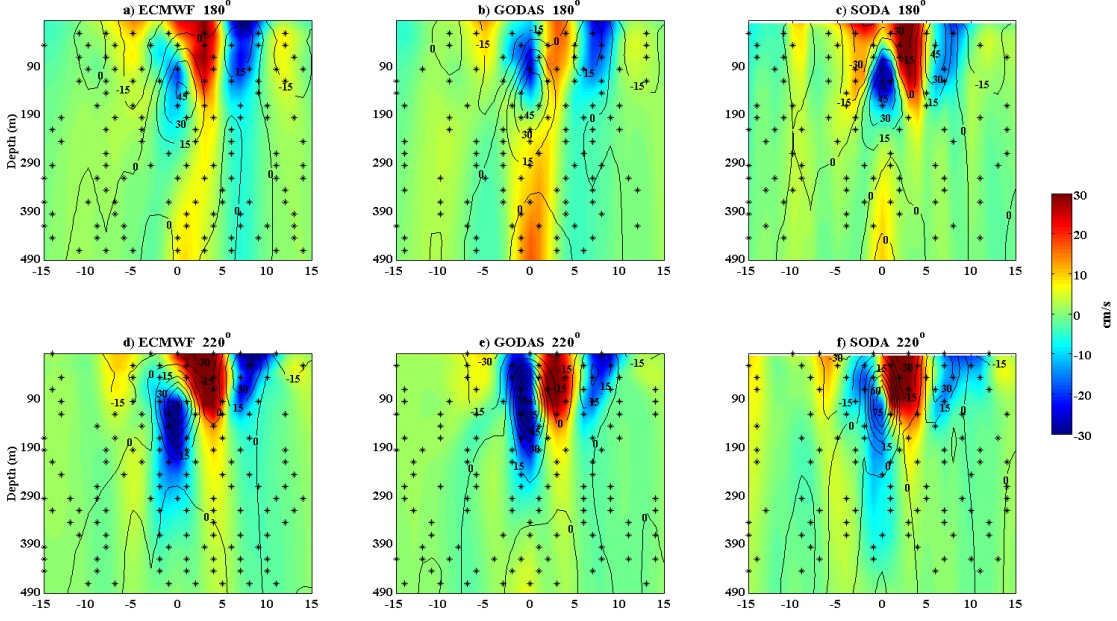

**Figure 2: Climatology (contour) and anomalies of ocean zonal current (shaded) on two profiles (Upper panels:**
**180 ̊, Lower panels: 220 ̊). Dotted regions are at 95% confidence level. Units: cm/s**
**4 Model results**
Under the extra 10% precipitation forcing in the tropical Pacific Ocean, SST shows significant decreasing
pattern over the Northern Hemisphere, especially over the northwest Pacific Ocean and the Northern





Atlantic Ocean as shown in Fig.3(1$^{st}$ EOF). The first mode accounts for 33% of the total variance. The
second mode accounting for 17% of the total variance is characterized by an El Nino-like pattern that the
spatial distribution of SST anomalies is opposite in the western and eastern Pacific Ocean. Power
spectrum method is used to capture the main features of the second principle component.   The power
spectrum peaks at 5 and 3.2 years, showing significant interannual variability. It is noted that the spatial
distribution of the second mode is similar with the composite maps of SST anomalies in Fig.1, confirming
the El Nino-like response of SST anomalies.
Four profiles are select to analyze the response of tropical ocean currents to the extra 10% precipitation
forcing. Profiles along 180° and 220° are used to represent the Pacific Ocean, and the 60° and 330°
profiles represent the Indian Ocean and the Atlantic Ocean respectively. Compared with the observation
results, main features of the tropical Pacific Ocean currents are well simulated by CESM (Fig.4a/b and
Fig.2). Previous studies have shown that the most significant changes of currents occur in the Pacific
Ocean. For the 180° profile, the South Equatorial Current weakens up to 8cm/s. The Equatorial Counter
Current shows negative anomalies on its north flank and significant positives anomalies on the upper part
of the Equatorial Latent Current, which results in the upward-tilting thermocline. For the 220° profile, the
South Equatorial Current and the upper part of the Equatorial Latent Current weaken, and the lower part
of the Equatorial Latent Current strengthen, which results in the thermocline deepening. Based on in-situ
measurements, it has been found that precipitation over the tropical Pacific Ocean east of 160° increases
during the 1997-1998 El Nino, while the ascending branch of the Walker circulation migrates eastward
and the Intertropical Convergence Zone (ITCZ) and South Pacific Convergence Zone (SPCZ) migrate
equatorward (Delcroix et al., (2002)). These changes are also accompanied by anomalous energetic
eastward surface currents along the equator, which is consistent with the results in this paper.
Comparatively speaking, currents in the Indian Ocean show little changes, and south subtropical ocean is
the area of most remarkable changes. However, the most remarkable changes of the Atlantic Ocean locate
in high latitudes (not shown), where the Gulf Stream weakens while the West Wind Drift strengthens.

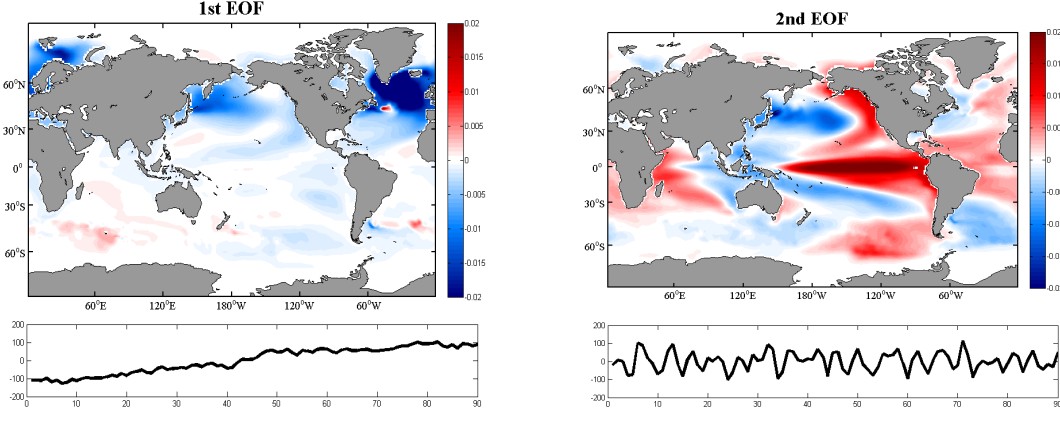

**Figure 3: Spatial and temporal distribution of the first two modes of SST anomalies between the sensitivity**
**experiment and the control experiment**
The model largely reproduces the currents characteristics shown in Section.3, especially for the 180°
profile, where the South Pacific Current and the upper part of the Equatorial Latent Current weaken
significantly. However, the positive anomalies lie between 150-300m depth reflecting the strengthening of




the Equatorial Latent Current is quite different from the composite results shown in Fig.2. For the 220°
profile, the equatorial counter current is not obvious. The differences between the model results and the
composite results are mainly due to the intensity of precipitation. The composite analysis is conducted for
the extreme precipitation conditions while the model simulates the response to the 10% precipitation
intensification. It reveals that precipitation in the tropical Pacific Ocean can not only influence the
magnitude of zonal currents, but also its direction especially over the east Pacific Ocean.

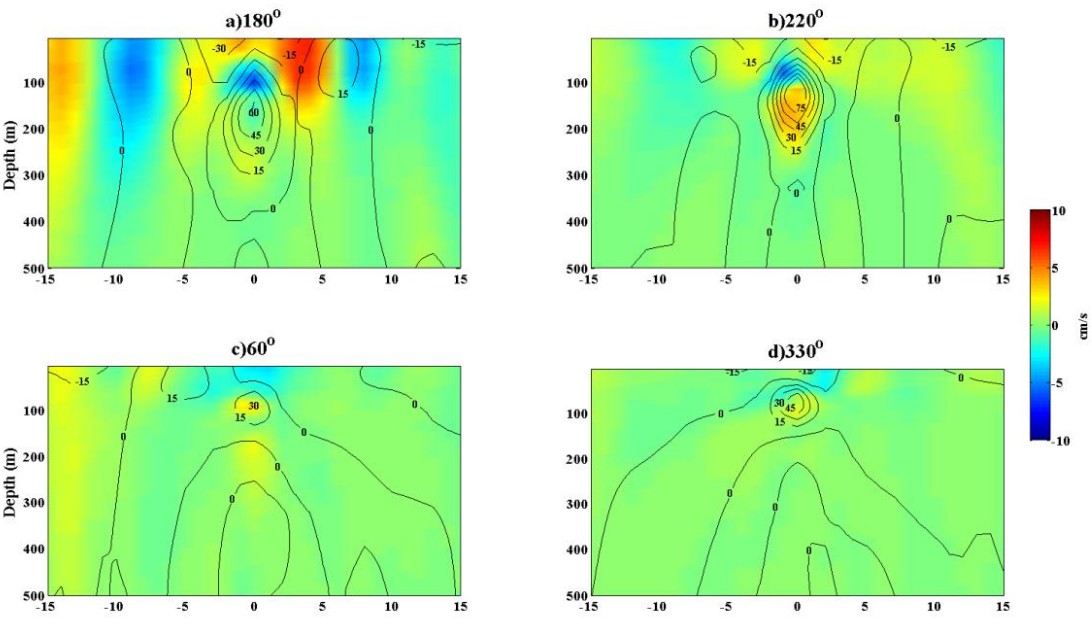

**Figure 4: Climatology and anomalies of ocean zonal current on four different profiles (a: 180°; b: 220°; c: 60°;**
**d: 330°. Contours represent the vertical climatology distribution of zonal current of the control experiment,**
**while the shading area represents the current anomalies. Unit:cm/s)**
Strong precipitation anomaly in the tropical Pacific Ocean could also cause significant change remotely.
The atmospheric circulation anomaly in equilibrium state is most significant around the Greenland Island
and well correlates with the sea surface temperature anomaly (SSTA) in Nino3 region. 500hpa
geopotential height anomalies are compared with the NINO3 SSTA, their annual correlation coefficient is
0.32 while it's 0.4 with a 7 years low-pass filtering. It shows that the El Nino-like SSTA could stimulate
atmospheric circulation anomaly in the North Atlantic high latitudes region, reflecting a fast process
connecting the tropical Pacific Ocean and the sub-Arctic region (Jevrejeva et al., 2004). It has been
confirmed that changes in tropical heating could trigger remote atmospheric response over the North
Atlantic Ocean, which changes the SST and sea ice in this region (Hoerling et al., 2001). Selten et al.,
(2004) studies the relationship between intensified tropical precipitation and the North Atlantic Oscillation
(NAO) trend under the circumstance of global warming. They found that intensified precipitation could
change the extra-tropical circulation in winter, which is characterized by a wave train encompassing the
whole Northern Hemisphere. The concept of this wave train, which is named as Circumglobal Waveguide
Pattern (CWP), is first brought forward by Branstator et al., (2002).





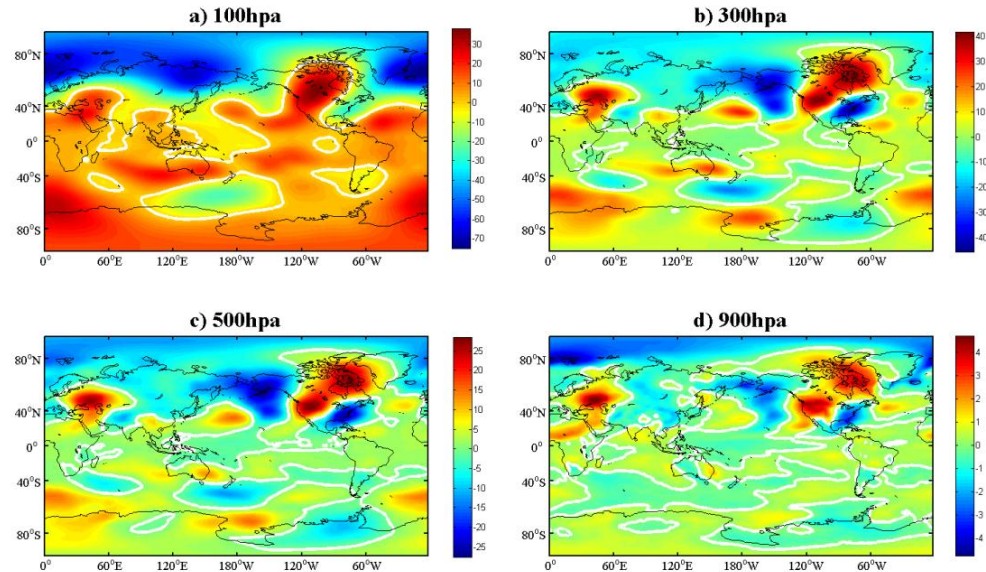

**Figure 5: Geopotential height anomalies in boreal winter averaged over the first ten years.**

**a):100hpa;b):300hpa;c):500hpa and d)900hpa. (White lines are contours for zero, Units: m)**

The geopotential height anomalies in winter averaged over the first ten years of the simulation are used to analyze the response of atmosphere circulation to precipitation anomaly in the tropical Pacific Ocean. It shows that the anomalies strengthen from lower atmosphere to upper atmosphere, which is characterized by a barotropic distribution (Fig.5). Spatial distribution of the 300hpa geopotential height anomalies shows a CWP pattern, which resembles the wave-train pattern in Selten et al., (2004). Branstator et al., (2002) indicate that the tropical ocean between 150°E and 180°E is easier to drive the CWP, which coincide the location with the water-hosing region in their sensitive experiment.

The negative and positive atmosphere anomalies are in the east part of the Greenland Island and in the Atlantic subtropical high region respectively, which resembles the positive phase of NAO. To further study the result of atmospheric circulation anomaly, sea ice extent and ocean-ice freshwater are examined. The sea ice extent in the Arctic Ocean exhibits clear seasonality with the maximum and minimum occurring in the boreal winter and boreal summer respectively. In summer, the sea ice extent exceeds 60%, which mainly locates around the North Pole, close to the Greenland Island and the Canadian Archipelago. In winter, the whole Arctic Ocean except the west part of the Europe is covered by thick sea ice. As for the response in the sensitive experiment, sea ice extent decreases up to 10% in summer due to the increasing SST in the Pacific Arctic. In the meantime, sea ice extent increases north of the Western Siberia and on both sides of the Greenland Island. The regions with most significant sea ice increasing locate in the Barents Sea and the south part of the Greenland Island (Fig.6). The variable named 'fresh' in Fig.7 represents the freshwater entering into the ocean melted from the sea ice. In summer, fresh changes are mainly along the sea ice, especially north of the Western Siberia and seas surrounding the Greenland Island. The changes are significant on the south part of the Greenland Island in winter. The freshwater along the sea ice edge in summer are accumulated from the Barents Sea. Under the advection effect of the




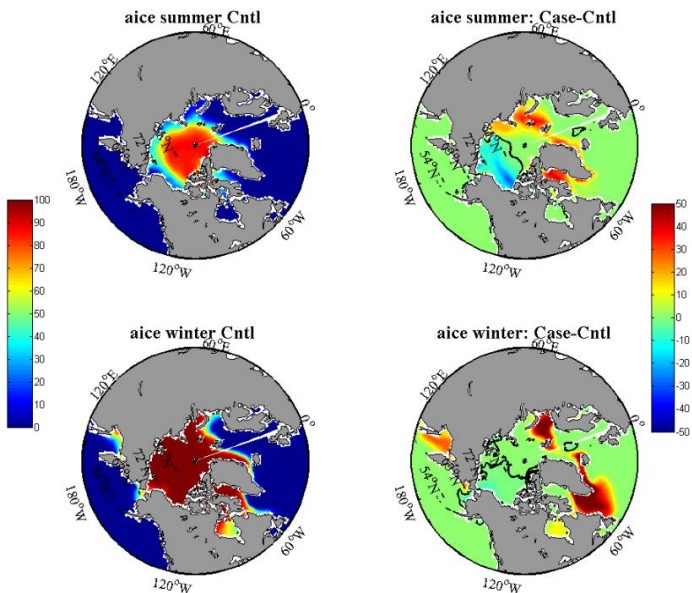

**Figure 6: Seasonal climatology and anomalies of sea ice extent(Units:%)**

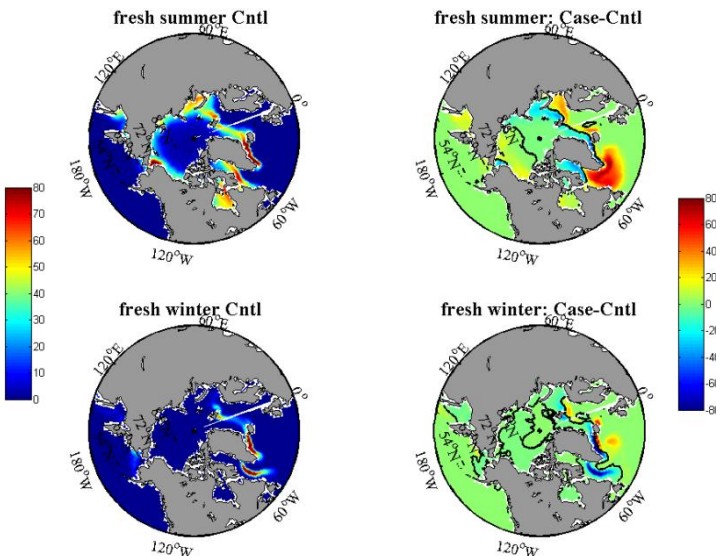

**Figure 7: Seasonal climatology and anomalies of sea ice extent(Units: cm/day)**
Transpolar Drift, it enters the north part of the Atlantic Ocean following the East Greenland flow and
makes the water in that region cool and fresh. Furthermore, a large amount of sea ice could be transported





to the south part of the Greenland Island by the East Greenland flow. The sea water thus gets cooler and fresher when the sea ice is melted. The freshwater in the high latitudes of the North Atlantic Ocean could reduce the mix layer depth and strengthen stratification by lightening the seawater and intensifying the buoyancy flux of the upper ocean. In turn, the strengthened stratification weakens the mixing and entrainment effect, which constrains the vertical heat exchange resulting in the temperature decrease. These processes form a significant positive feedback mechanism.

## 5 Summary

In this study, historical datasets are used to evaluate the characteristics of ocean currents and SST during the strong precipitation periods using composite analysis method. It is found that both the ocean current and SST encounter significant changes during the strong precipitation period. Strong precipitation over the tropical Pacific Ocean could trigger an El Nino-like SSTA response, with the positive SSTA anomalies in the east tropical Pacific Ocean and negative SSTA anomalies in the west tropical Pacific Ocean. Moreover, the tropical Pacific currents in the three different datasets of this study show a consistent response to freshwater anomalies on the tropical Pacific Ocean.

A series of experiments using CESM is performed to study the response of Pacific Ocean to an extra 10% precipitation forcing. The model results show that the intensified freshwater in the tropical Pacific Ocean could generate a significant cooling tendency in the northwest Pacific Ocean and the Northern Atlantic Ocean, and trigger an El Nino-like warming in the tropical Pacific Ocean. For the 180° profile, currents change similarly between the model and composite results. However, for the the 220° profile, it is opposite to the lower part of the Equatorial Latent Current. It reveals that precipitation in the tropical Pacific Ocean can not only influence the magnitude of zonal currents, but also its direction especially over the east Pacific Ocean. Through a wave train encompassing the whole Northern Hemisphere named as CWP, the North Atlantic atmospheric circulation responds to the freshwater anomalies in a NAO-like pattern.. The anomalous atmospheric circulation could transport more freshwater (water and sea ice) from the ice edge to the North Atlantic Ocean. The sea ice melts in summer and freshen the upper ocean, which makes the ocean more stable. It thus constrains vertical heat transport and makes the upper water cooler, forming a significant positive feedback mechanism.

### Data availability

Reanalysis datasets used in this study (GPCP, ECMWF, GODAS and SODA) are all obtained from ASIA-PACIFIC DATA-RESEARCH CENTER (apdrc), with the website address http://apdrc.soest.hawaii.edu/las/_v6/dataset?catitem=0

### Acknowledgement

This study is supported by the Public science and technology research funds projects of ocean (201505013), NSFC project Nos. 41376008, 41376016, 41576029 and 41106024.



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
