# Peer review of "Sensitivity study of the tropical Pacific precipitation anomalies"

_Natural Hazards and Earth System Sciences, 2016_

## Referee Comment (RC1) · Anonymous Referee #1 · 11 Apr 2016

Review of manuscript nhess-2016-83: "Sensitivity study of the tropical Pacific precipitation anomalies" by Shouwen Zhang, Hua Jiang, Hui Wang, Ling Du and Dakui Wang

Originality: Fair Technical Quality: Fair Clarity of Presentation: Fair Significance: Fair

This work studied the effect of an enhanced tropical Pacific precipitation on the tropical Pacific SST, tropical upper ocean current, the atmospheric circulation in the Northern Hemisphere and the sea ice. My first impression on this paper is "what a mess!" In this short manuscript, the authors tried to describe the comprehensive changes in the Earth climate, involving El Nino, CWP, NAO, Arctic sea ice, and on seasonal, interannual and longer-timescale. All these things are mixed in this manuscript, making this paper lack of focus. This manuscript has serious logical problem and technical default. There are also no convincing physical explanations on how and why the tropical precipitation
change can cause those consequences. Therefore, I recommend major revision of this manuscript. My major comments are listed below.

Major comments:

1. All figures are badly demonstrated. The color bar used in the figures are not clear to show the positive/negative changes in those variables, particularly in Figs. 4-7. The figure captions are also not specific at all. For example, in Fig. 1, what are the strong and weak period? In Fig. 2, the anomalies from what? In Fig. 3, how the 2nd EOF mode is obtained? And so on, all the figure captions are vague!

2. Several important figures are not available. For example, what are the strong/weak precipitation patterns in the tropical Pacific in the reanalyzed data, and the corresponding SSTA pattern from the climatological mean? Fig. 1 actually shows the El Nino pattern, which might have nothing on the precipitation.

3. The serious logical problem in this paper, I think, is the causality between the SST and precipitation. It is undoubted that changing precipitation in a coupled system can result in big responses in other climate variables. However, based on observational data or reanalysis data, the authors should have a basic sense on the lag/lead relationship between SSTA and precipitation. The precipitation is pretty much an internal variable of climate system with huge uncertainty.

4. What is the strong precipitation pattern over the tropical Pacific in the coupled model?

5. The authors provided almost no physical explanations on how the precipitation causes a series of consequences.

6. The so-called CWP is not clear at all.

7. The authors think "strong precipitation trigger an El Nino-like SSTA", based on Fig.

3. El-Nino mode is the dominant mode in the tropics in any case (in both control run and sensitivity experiment), which cannot be attributed to the enhanced precipitation.
Instead, the author should compare the El Nino mode in CTRL and experiment and see how the precipitation affect this mode.

8. In general, the authors should rewrite this paper completely and make clear what timescale and what regions they focus on. To my opinion, in this short paper, the authors should focus on the mean climate change in response to the enhanced precipitation and focus on the tropical upper ocean. This is no need to do EOF and discuss CWP, NAO, seasonal sea ice and so on. By the way, in such a short modeling duration (90 years), the upper ocean changes are not curtain at all to be attributed to the surface rainfall. 10% precipitation enhancement is not strong enough to cause those significant change globally.

---

## Referee Comment (RC2) · Anonymous Referee #2 · 13 Apr 2016

This paper investigated the effect of an increased tropical Pacific precipitation on the tropical Pacific SST, upper layer oceanic current, the atmospheric circulation in the Northern Hemisphere and the sea ice. The response of global ocean and atmosphere to precipitation anomalies in the tropical Pacific Ocean is a important topic. This paper reports some interesting results on the topic, but some comments below should be taken into account by authors of this paper, and I recommend major revision of this manuscript.

1) Precipitation correlates significantly with SST over the tropical Pacific Ocean, as shown in Fig.1in the paper, but this correlation is only a statistical result. This correlation cannot support the conclusion that the spatial distribution of temperature anomalies caused by strong precipitation resembles with the SST anomalies characteristics during El Nino periods. On the contrary, the precipitation anomalies may be caused by

the SST anomalies.

2) The design of the numerical experiment is not appropriate. Precipitation involves many atmospheric physical and dynamical complex processes, it is not a proper method to add 10% of precipitation on each time-step in the coupler. The better choice maybe is to run the CESM model in RCP 4.5 or 8.5 scenarios, and then analyses the output of the model.
* * *